# Effects of Solid-State Fermented Wheat Bran on Growth Performance, Immune Function, Intestinal Morphology and Microflora in Lipopolysaccharide-Challenged Broiler Chickens

**DOI:** 10.3390/ani12091100

**Published:** 2022-04-24

**Authors:** Jishan An, Jingjing Shi, Kuanbo Liu, Aike Li, Beibei He, Yu Wang, Tao Duan, Yongwei Wang, Jianhua He

**Affiliations:** 1College of Animal Science and Technology, Hunan Agriculture University, Changsha 410128, China; ajs@stu.hunau.edu.cn; 2Academy of National Food and Strategic Reserves Administration, Beijing 100037, China; sjj@ags.ac.cn (J.S.); lkb@ags.ac.cn (K.L.); lak@ags.ac.cn (A.L.); hbb@ags.ac.cn (B.H.); bright0224@126.com (Y.W.); dt@ags.ac.cn (T.D.)

**Keywords:** fermented wheat bran, lipopolysaccharide, growth performance, immune function, broiler

## Abstract

**Simple Summary:**

Fermented wheat bran was produced by solid-state fermentation method, and the changes of nutritional indexes before and after solid-state fermentation were analyzed. Then, Arbor Acres (AA) broilers were used to study the effects of fermented wheat bran on growth performance, serum immune function, intestinal morphology and microflora of broilers under bacterial lipopolysaccharide stress. The present results indicated that the nutritional value of wheat bran was improved by solid-state fermentation, and wet FWB improved the immune profile of broiler chicken under stress.

**Abstract:**

The study evaluated the effects of dry and wet solid-state fermented wheat bran (FWB) on growth performance, immune function, intestinal morphology and microflora in lipopolysaccharide (LPS)-challenged broiler chickens. The experiment was designed as a 2 × 3 factorial arrangement. A total of 252 one-day-old Arbor Acres male broiler chickens were randomly allocated to 1 of 6 treatments: basal diet + sterile saline (negative control, NC), basal diet + LPS (positive control, PC), 7% dry FWB + sterile saline (FWB-I), 7% dry FWB + LPS (FWB-II), 7% wet FWB + sterile saline (FWB-III) and 7% wet FWB + LPS (FWB-IV), with containing 6 replicate cages/treatment and 7 broiler chickens/cage, and the experimental period lasted for 42 days. Broilers were intraperitoneally injected with either 0.5 mg LPS or sterile saline solution per kg body weight at 16, 18 and 20 d of age. Growth performance, serum immunological parameters and indicators related to intestinal health were analyzed on days 21 and 42. Compared with NC, dry and wet FWB significantly increased (*p* < 0.05) average daily feed intake of days 21 to 42, and increased (*p* < 0.05) the villus height and villus height to crypt depth ratio of ileum on day 21, decreased (*p* = 0.101) the jejunum crypt depth and decreased (*p* < 0.05) the *Lactobacillus* and *Bifidobacterium* counts of the cecum digesta on day 42. Compared with NC, FWB-II and FWB-IV significantly increased (*p* < 0.05) the levels of serum total protein and globulin on day 21; compared with the basal diet groups, dry and wet FWB groups significantly increased (*p* < 0.05) glucose levels on day 21, and wet FWB significantly decreased (*p* < 0.05) alanine aminotransferase levels on day 42. Compared with PC and FWB-II, FWB-IV significantly increased (*p* < 0.05) the level of serum immunoglobulin G on day 21. Compared with PC and FWB-II, FWB-IV significantly decreased (*p* < 0.05) the levels of serum pro-inflammatory cytokines interleukin (IL)-6, IL-8, IL-1*β* and acute C reactive protein (CRP) on day 21; compared with FWB-III, FWB-IV significantly decreased (*p* < 0.05) the levels of IL-6, IL-8, CRP and tumor necrosis factor alpha on day 42, but the levels of IL-4 and IL-10 were significantly increased (*p* < 0.05) on days 21 and 42. These results indicated that supplementing 7% dry or wet FWB can improve growth performance and serum immune functions of broilers, which effectively alleviate the LPS-challenged damage, and wet FWB had a better effect than dry FWB.

## 1. Introduction

In recent years, the immune stress caused by external pathogens seriously inhibit the growth potential of broiler chickens [1,2]. The endotoxin lipopolysaccharide (LPS) is a major component of the cell wall of Gram-negative bacteria. LPS challenge by orally or abdominally injection can cause an acute inflammatory response, increase the release of reactive oxygen species in animals and cause obvious pathological changes [3,4]. Therefore, the development of effective strategies to control bacterial infection in the poultry industry is very essential. At present, antibiotic-free feeds are being used in most countries. Microbial fermented feed is one of the possible strategies to improve the functional value of feed ingredients [5,6] by increasing the feed utilization rate, maintaining the balance of intestinal microflora [7] and producing bioactive and beneficial metabolites [5,8]. Jazi et al. [9] reported that the basal diet supplemented with fermented soybean meal significantly improved growth performance, lowered *Salmonella* colonization and reduced heterophil to lymphocyte ratio; the functional compounds in fermented soybean meal, such as *Lactobacillus plantarum*, organic acid, short chain fatty acid and digestive enzymes, improved the intestinal morphology characteristics and immune response of young broiler chickens infected by *Salmonella typhimurium*. Similarly, Ashayerizadeh et al. [10] reported that fermented rapeseed meal significantly reduced the colonization of *Salmonella typhimurium* in the gut, decreased the heterophil to lymphocyte ratio and improved the growth performance of broiler chickens challenged by *Salmonella typhimurium*. Thus, fermented feed can be used as an alternative to antibiotics in animal production.

Wheat bran (WB) as the main byproduct of wheat starch processing, has rich nutrients and bioactive compounds [11,12], such as phenolic components (mainly phenolic acids, flavonoids, and lignans), soluble dietary fiber and polysaccharides. Adding a proper proportion of wheat bran to animal diets can promote the development of intestinal function and growth performance, but excessive addition can cause negative impact for its high lignocellulose content, non-starch polysaccharides, phytic acid and mycotoxin risk [7,13]. Several studies have emphasized that the solid-state fermentation (SSF) technology, as an effective approach for improving the nutritional quality of by-products, can effectively improve the nutritional value, physical and flavor properties of wheat bran [14,15,16]. Zhao et al. [17] found that fermentation of wheat bran by *Saccharomyces* and *Lactobacillus* increased the content of water extractable arabinoxylan by 3–4 times, increased the content and bioavailability of soluble dietary fiber and total free phenolic, and degraded over 20% of phytic acid. Additionally, adding 5% dry fermented wheat bran (FWB) to broilers (Ross 308) diets improved the growth performance, increased the *Lactobacillus* counts in ileum, and maintained the beneficial intestine environment [18]. Zhang et al. [19], Lee et al. [20] and Teng et al. [21] found that the solid-state fermented wheat bran had promoting effects on growth performance, nutrient digestibility and anti-inflammation in broiler chickens. The drying process not only increased the cost of fermented feedstuffs, but reduced the contents of volatile substances and probiotics, so direct application of wet fermented feed could reduce the nutrient loss and processing cost. Previous studies mainly focused on the application of dry fermented wheat bran, whether the application of wet fermented feed had better effects than dry fermented feed needs further study, especially in broiler chickens challenged with LPS. Therefore, this study aims to determine the bioactive components in solid-state fermented wheat bran and compare the effects of dry and wet FWB on growth performance, immune function, intestinal morphology and microflora in LPS-challenged broiler chickens.

## 2. Materials and Methods

### 2.1. Microorganisms and Inoculum Preparation

*Lactobacillus plantarum* (LP17-1) and *Bacillus subtilis* (BS17-1) were obtained from Academy of National Food and Strategic Reserves Administration. *Aspergillus fragrant* (AF) was purchased from Wuhan Jiacheng Biotechnology Products Co., Ltd. (Wuhan, China). *Saccharomyces cerevisiae* (SC) was purchased from the Angel yeast Co. (Hubei, China). LPS from *Escherichia coli* (*E. coli*) serotype O55:B5 (L2880) was purchased from Sigma-Aldrich Chemical Co. (Saint Louis, MO, USA). The strains were prepared by shaking flask cultured with MRS broth, lysogeny broth and yeast peptone dextrose broth, respectively. Briefly, a 250 mL Erlenmeyer flask filled with 100 mL of different inoculum was covered with tin foil and autoclaved at 121 ± 1 °C for 15 min before use. Different microorganism was transferred to the related agar plate and cultured at 37 °C for 24 h. The concentration of viable bacteria in suspension was detected by the plate count method [8].

### 2.2. Preparation of Solid-State Fermentation WB by Microorganism

Solid-state fermentation of wheat bran was performed in a facultative anaerobic solid-state fermentation equipment to produce FWB. Wheat bran was mixed with 80% sterile water, and added to a 10% mixture inoculum (3% broth culture of LP, 2% broth culture of BS, 4% broth culture of SC and 1% broth culture of AF), 2% glucose, then fermented at 27–33 °C for 3 days and stirred regularly per day. FWB was dried by boiling fluidized bed at 55 °C to collect dry FWB. Both dry and wet FWB samples were sealed and stored at 4 °C to determine the physical and chemical parameters.

### 2.3. Extraction and Determination of Bioactive Compounds of FWB

Wheat bran and wet FWB samples were freeze-dried and crushed through 40-mesh sieve. The total phenolic content was determined using the Folin–Ciocalteu reagent described by Wang et al. [22]. Briefly, 1 mL of FWB extract was pipetted into a test tube with a stopper, then added 5 mL of 10% Folin reagent, and after reaction at room temperature for 5 to 8 min, added 4 mL of 7.5% Na_2_CO_3_ solution and vortexed to mix thoroughly. After reaction for 60 min, the absorbance was measured by UV-spectrophotometer at 765 nm, and an equation was obtained from the standard gallic acid (GA), standard graph was used to determine the phenolic compounds of each sample (mg of GA equivalent per g DW of sample). The crude polysaccharide content was quantified by a modified phenol-sulfuric acid method [8]. Briefly, 1 mL of polysaccharide solution was pipetted into a 10 mL test tube with stopper, then added 1 mL of 5% phenol solution and 5 mL of concentrated sulfuric acid solution, and the reaction mixture with extracts were kept in a 30 °C water bath for 20 min, the absorbance was measured by UV-spectrophotometer at 490 nm after cool to room temperature. The result was expressed as mg of the glucose equivalent/g DW using the standard glucose calibration cure. The total dietary fiber was evaluated using the Enzyme-weight method (Fibertec 1023, FOSS, Hillerod, Denmark) [21]. The mannan content was determined by pre-column derivatization HPLC [23]. Briefly, 400 μL of FWB and WB extract, 0.5 mol/L PMP methanol solution, and 0.3 mol/L sodium hydroxide solution were pipetted into a 10 mL centrifuge tube with stopper, then derivatized in a 70 °C water-bath for 90 min after mixing well, added 500 μL 0.3 mol/L HCL after cool, mixed again and washed with 2 mL chloroform solution 4 times, centrifuged and filtered through a 0.45 μm membrane filter, then the supernatant sample was determined by HPLC. The standard curve and regression equation were obtained from the standard mannose, with the concentration of the standard sample as the abscissa and the peak area as the ordinate, which was used to determine the mannan content in each sample.

### 2.4. Animals and Management

The animal experiment was approved by the Animal Care and Use Committee of Academy of National Food and Strategic Reserves Administration (registration number: 2018L02). The animal experiment was performed at the Wuqing base of Academy of National Food and Strategic Reserves Administration (Tianjin, China).

A total of 252 healthy one-day-old Arbor Acres male broilers (average body weight 43.8 ± 0.15 g) were purchased from a commercial hatchery (Huadu Yukou Co., Ltd. Beijing, China), and randomly allocated to 1 of 6 treatments: basal diet + sterile saline (negative control, NC), basal diet + LPS (positive control, PC), 7% dry FWB + sterile saline (FWB-I), 7% dry FWB + LPS (FWB-II), 7% wet FWB + sterile saline (FWB-III) and 7% wet FWB + LPS (FWB-IV), with containing 6 replicate cages/treatment and 7 broiler chickens/cage. The experiment was designed as a 2 × 3 factorial arrangement with LPS treatments (injected with either 0.5 mg LPS or saline per kg body weight) and diet treatments (basal diet, 7% dry FWB and 7% wet FWB). LPS was dissolved in sterile saline to prepare a concentration of 0.5 mg/mL. At 16, 18 and 20 d of age, birds were received an intraperitoneal administration of LPS or an equal amount of saline (0.5 mL).

The animal experiment was conducted in three phases, including an early period (days 1 to16), LPS-challenged period (days 17 to 21) and later period (days 22 to 42). All chickens were housed in a temperature-controlled room with continuous light and breeding in (0.8 × 0.7 m^2^). Birds were offered feed and water ad libitum. The temperature was maintained at 32 to 35 °C during the first week, which was progressively reduced by 2 to 3 °C a week until it reached 25 °C. The relative humidity was set at 50% throughout the trial. All chickens were vaccinated with an inactivated Newcastle disease-infectious bronchitis vaccine on day 7 and with infectious bursal disease vaccine on day 14. A starter diet (day 1 to 21) and grower diet (day 22 to 42) were designed to meet the nutrient requirements of China feeding standard of chicken (NY/T 33-2004) (Table 1). All feeds were free of antibiotics. The wet fermented wheat bran diet was sealed, the moisture, mycotoxin and smell were evaluated before feeding, to ensure the wet fermented wheat bran diet was in good condition.

### 2.5. Sample Collection and Procedures

On days 16, 21 and 42 of the experiment, one bird per replicate cage from each treatment group (total of 6 birds/treatment) was randomly selected and the body weight of all birds were recorded after an 8 h feed withdrawal. The growth performance was evaluated by final body weight (FBW), average daily gain (ADG), average daily feed intake (ADFI) and feed conversion ratio (feed/gain, FCR), which was recorded on a cage basis during different periods.

On days 21 and 42, blood samples were obtained in 5 mL vacutainer tubes from the brachial vein of birds, then centrifuged at 3000× *g* for 15 min. Sera samples were separated and stored at −20 °C until biochemical analysis. Then the birds were euthanized by exsanguination and abdominal cavities were opened. The cecum digesta were collected in sterile EP (Eppendorf) tubes, then dipped immediately in liquid nitrogen and kept at −80 °C. Furthermore, 1 cm of the middle segments of jejunum and ileum were collected and fixed in 4% paraformaldehyde solution for later morphometrical assays.

### 2.6. Measurement of Serum Biochemical Parameters

The serum biochemical parameters, including glucose (GLU, hexokinase method), triglyceride (TG, GPO-PAP method), cholesterol (TC, GPO-PAP method), alanine aminotransferase (ALT, alanine substrate method), aspartate aminotransferase (AST, aspartate substrate method), total protein (TP, biuret method), albumin (ALB, bromocresol green method) and globulin (GLB), which were determined by commercially available assay kits (Biosino Biotechnology and Science Inc., Beijing, China) and performed using an automatic biochemical analyzer (Mairui BS-420 autoanalyzer, Shenzhen, China). All procedures were performed according to the manufacturer’s instructions.

### 2.7. Determination of Immunoglobulins and Cytokines

Serum immunoglobulin G (IgG), immunoglobulin A (IgA) and immunoglobulin M (IgM) were determined by immunoturbidimetry; interleukin-4 (IL-4), interleukin-6 (IL-6), interleukin-8 (IL-8), interleukin-10 (IL-10), interleukin-1 beta (IL-1*β*), tumor necrosis factor alpha (TNF-α) and acute C reactive protein (CRP) were determined by an ELISA method. ELISA kits were purchased from Beijing SINO-UK Institute of Biological Technology (Beijing, China). All procedures were performed according to the manufacturer’s instructions.

### 2.8. Intestinal Morphology and Microbial Counts in Cecum Digesta

The paraformaldehyde-fixed gut wall samples were washed in phosphate-buffered saline solution, then dehydrated by ascending concentrations of ethyl alcohol (70% to absolute alcohol), cleared in xylene and embedded in paraffin wax. Sections of 4–5 μm thickness were stained with hematoxylin-eosin for general morphometry, according to Fischer et al. [24]. The samples were analyzed by light microscopy (Eclipse Ci-L, Nikon, Japan), and software (Image-Pro Plus 6.0) was used to measure the villus height (Vh) and crypt depth (Cd) in 25 favorably-oriented and representative samples per treatment. The ratio of villus height to crypt depth (V/C) was also calculated.

One gram of digesta sample was collected from the cecum. Samples were serially diluted with 9 mL phosphate-buffered saline solution and mixed thoroughly for enumeration of microbial populations. *E.coli* was cultured with Aobox EMB agar under aerobic conditions at 37 °C for 48 h. Strains of *Lactobacillus* and *Bifidobacterium* were cultured with Aobox MRS agar and BS agar, respectively. The MRS agar plates and BS agar plates were incubated under anaerobic conditions at 37 °C for 48 h. The microflora numbers were calculated by plate count method, and expressed as log10 CFU per gram of cecum digesta [25].

### 2.9. Statistical Analysis

The data was analyzed using SAS software (version 8.02, SAS Institute Inc., Cary, NC, USA). A student T-test was used to assess the effect of the fermentation process on the bioactive compounds and chemical composition of wheat bran. Data related to feeding period of broiler chickens were analyzed in a 2 × 3 factorial arrangement design using the GLM procedures of SAS software. The differences among treatments were evaluated by Duncan’s multiple range tests, and values of *p* < 0.05 were considered statistically remarkable.

## 3. Results

### 3.1. Nutrients and Bioactive Contents in Fermented Wheat Bran

The nutrient contents of WB and FWB are presented in Table 2. Based on dry matter basis, compared with WB, the contents of crude protein, acid-soluble protein and gross energy in FWB were significantly increased by 19.89% (*p* < 0.05), 248.29% (*p* < 0.05) and 8.32% (*p* < 0.05), respectively. The contents of crude fat and crude fiber were reduced by 39.41% (*p* < 0.05) and 7.73% (*p* < 0.05), respectively.

The major functional bioactive compounds are presented in Table 3. Compared with WB, the contents of total polyphenol, soluble dietary fiber, total dietary fiber and mannan were significantly increased by 207.33% (*p* < 0.001), 121.23% (*p* < 0.05), 58.77% (*p* < 0.001) and 95.65% (*p* < 0.05), respectively. The content of crude polysaccharide was increased by 78.88%.

### 3.2. Growth Performances

The results of the growth performance of broiler chickens are presented in Table 4. There was no significant difference in growth performance between the same diet groups at early period (days 1 to 16). Compared with basal diet groups, dry and wet FWB groups significantly increased (*p* < 0.05) ADG and ADFI of broilers, while FCR was not significantly affected (*p* > 0.05) on day 16. During the LPS-challenged period (days 17 to 21), compared with NC, FWB-III significantly increased (*p* < 0.05) ADFI of broilers, all other parameters were statistically similar for all other treatments. During the later period (days 22 to 42), compared with NC, FWB-I and FWB-III significantly increased (*p* < 0.05) ADFI and FCR.

### 3.3. Intestinal Morphology

The results of the intestinal morphology of broiler chickens are presented in Table 5. During the LPS-challenged period (days 17 to 21), compared with the saline groups, the Vh, Cd and V/C of jejunum and ileum were not significantly affected (*p* > 0.05) in the LPS-challenged groups. Compared with PC, FWB-II and FWB-IV significantly increased (*p* < 0.05) the Vh and V/C in ileum. There was no significant difference (*p* > 0.05) between the dry and wet FWB groups. During the later period (days 22 to 42), the dry and wet FWB had a tendency to reduce (*p* = 0.101) the Cd of the jejunum.

### 3.4. Microbial Count in Cecum

The results of the cecum microflora numbers of broiler chickens are presented in Table 6. During the LPS-challenged period (days 17 to 21), compared with the saline groups, the *E. coli* counts were significantly increased (*p* < 0.05) in LPS-challenged groups, but had no effect (*p* > 0.05) on the *Bifidobacterium* counts. Compared with the NC, FWB-III significantly reduced (*p* < 0.05) the *Lactobacillus* and *Bifidobacterium* counts, while *E. coli* counts were not significantly affected (*p* > 0.05) on days 42. During the later period (days 22 to 42), the *E. coli* counts of the cecum digesta were not significantly affected (*p* > 0.05) among treatments. Compared with the PC, FWB-II and FWB-III significantly decreased (*p* < 0.05) the *Bifidobacterium* and *Lactobacillus* counts of the cecum digesta; while FWB-IV significantly increased (*p* < 0.05) the *Lactobacillus* counts and decreased (*p* < 0.05) the *Bifidobacterium* counts, while *E. coli* counts were not significantly affected (*p* > 0.05). There was no significant difference (*p* > 0.05) between dry and wet FWB.

### 3.5. Serum Biochemical Parameters

The results of the serum biochemical of broiler chickens are presented in Table 7, during the LPS-challenged period (days 17 to 21), there was no significant difference (*p* > 0.05) in serum biochemical parameters between the LPS groups and the saline groups. Compared with the NC, FWB-II and FWB-IV significantly increased (*p* < 0.05) TP and GLB levels. Compared with the basal diet groups, dry and wet FWB groups significantly increased (*p* < 0.05) GLU level. There was no significant difference (*p* > 0.05) between dry and wet FWB groups. During the later period (days 22 to 42), compared with the NC, FWB-III and FWB-IV significantly decreased (*p* < 0.05) the ALT level, while other serum biochemical parameters were not significantly affected (*p* > 0.05). Compared with the basal diet groups, the dry FWB groups had no significant difference (*p* > 0.05) in serum biochemical parameters, but the wet FWB groups significantly decreased (*p* < 0.05) ALT level.

### 3.6. Serum Immunoglobulins and Cytokines

The results of serum immunoglobulins and cytokines of broiler chickens are shown in Table 8 and Table 9, respectively. During the LPS-challenged period (days 17 to 21), the serum cytokine TNF-α and CRP were significantly increased (*p* < 0.05), but immunoglobulin level was not significantly affected (*p* > 0.05) in LPS-challenged broilers. Compared with the PC and FWB-II, the FWB-IV significantly increased (*p* < 0.05) the levels of IgG on day 21. Compared with PC, FWB-IV significantly decreased (*p* < 0.05) the levels of pro-inflammatory cytokines IL-6, IL-8 and CRP, and the levels of anti-inflammatory cytokines IL-4 and L-10 were significantly increased (*p* < 0.05). During the later period (days 22 to 42), there was no significant difference (*p* > 0.05) in serum immunoglobulins among the diet treatments. Compared with FWB-III, FWB-IV significantly decreased (*p* < 0.05) the levels of pro-inflammatory cytokines TNF-α, IL-6, IL-8 and CRP, but IL-4 and IL-10 were significantly increased (*p* < 0.05).

## 4. Discussion

It was reported that solid-state fermentation (SSF) is an effective approach for improving the nutritive value of grain and oil processing byproducts by reducing the cellulose content and increasing a variety of biological active substances [16,19]. The results of the present study revealed that the CF content of WB was significantly decreased after fermentation, and the content of CP and acid soluble protein was significantly increased. The decrease of CF content may be related to the cellulase by probiotic bacteria (lactic acid bacteria, bacillus subtilis) during fermentation. Besides, the increase of CP content was likely due to the fermented material concentration effect at the expense of carbohydrates loss in the process of microbial fermentation [19].

Phytogenic-derived phenolic compound and polysaccharide can exert growth-promoting effect by its antioxidant function [26]. He et al. [27] reported that diets supplemented with 350 or 500 mg/kg resveratrol (a phytoalexin polyphenolic compound) improved the growth performance of broilers under heat stress. Wang et al. [28] also found that astragalus polysaccharides had a better potential to improve the growth performance of broilers. The current results showed that the basal diet supplemented with FWB significantly improved the growth performance of broiler chickens in early period, which was consistent with the results of Zhang et al. [19]. Wheat bran contains 3–4% crude polyphenols, 20–35% crude polysaccharides and 2–6% soluble dietary fiber [22,29], which have potential physiological functions. Previous research had proved that solid-state fermentation of wheat bran by microorganisms could effectively increase bioactive compounds contents [30,31]. The present results also showed that the contents of total polyphenol, soluble dietary fiber, total dietary fiber and mannan were significantly increased. Thus, the increase in the bioactive compound content of FWB is one of important factors to improve the performance of broilers.

LPS is widely used to induce immune and oxidative stress models, which are characterized by anorexia, fever, impaired growth performance, and increased proinflammatory cytokine production for animals [32]. In the current study, LPS-challenged treatment decreased ADG and ADFI at 21 days of age, which proved that we successfully constructed the immune stress model by injection LPS in broilers [2]. The reason for growth inhibition was mainly ascribed to the partitioning of nutrients from growth to support inflammatory-related processes [2,33]. Kang et al. [30] reported that wheat bran had anti-inflammatory properties in LPS-challenged mice. The current results indicated that FWB significantly alleviated the growth inhibition induced by LPS, and the bioactive compounds in FWB had better effect for maintaining animal health under stress conditions. In addition, FWB-III and FWB-IV diets had no significant effect on growth performance of broilers compared to NC, which can be attributed to the improvement in nutritive value and palatability of wet FWB diets, resulting in the similar growth performance with basal diet [19,34]. In the small intestine, longer villi height and shallower crypt depth can provide a larger surface area for the digestion and absorption of nutrients [15,21]. The current results showed that the basal diet supplemented with FWB increased the Vh and V/C of broilers ileum after LPS-challenged. The improvement in intestinal morphology may be due to increase the expression of tight junction proteins in intestinal epithelial cells by wheat bran polysaccharide and free phenolic for protecting intestinal integrity [18]. It was reported that the soluble dietary fiber and mannan in wheat bran could help to optimize the composition of intestinal microorganisms, and produce beneficial effect on the growth performance of broilers [35,36]. The current results showed that the E. coli counts of cecum digesta were not significantly affected among treatments groups during the later period, whereas the Lactobacilli counts were significantly increased in FWB-IV. The reason may be that polysaccharides, mannan and probiotic bacteria (lactic acid bacteria, bacillus subtilis) in FWB changed the microbial structure in cecum, and produced more beneficial metabolites (organic acids) to inhibit the proliferation of pathogenic bacteria [7,37]. Serum biochemical parameters reflect the changes in absorption and metabolic function. The present results showed that there was no significant difference in TP, ALB, GLB, blood sugar, blood lipid levels and transaminase activity of broilers after LPS-challenged, which was inconsistent with the results of Wu et al. [38]. However, the levels of serum TP, ALB and GLB were significantly increased in FWB group. Possibly, bioactive compounds in FWB improved the intestinal integrity, and further promoted the digestion and absorption of protein and protein synthesis in the body [39]. Serum immunoglobulins play an important role in the regulation of inflammation. The present results showed that the levels of serum IgG, IgM and IgA were significantly affected after LPS-challenged, which was inconsistent with the results of Yang et al. [32]. The reason may be that the low concentration of IL-6/TNF-α induced by stress response promotes or induces the development of T and B lymphocytes, including maturation, differentiation and antibody generation, thus alleviating the humoral immunity disorder of the body. However, wet FWB significantly increased the levels of serum IgG and IgA in LPS-challenge broilers. Changes in the secretion levels of cytokines could accompany with the balance of pro-inflammatory cytokines (TNF-α, IL-6, IL-8, IL-1β and CRP) and anti-inflammatory cytokines (IL-4 and IL-10). Wu et al. [40] found that treatment with 0.5 mg/kg body weight LPS induced acute inflammation and resulted in increased contents of pro-inflammatory cytokines in the serum, indicating that a successful inflammation model was established, which was similar to our results. The current results showed that LPS-challenged treatment significantly increased the TNF-α concentration, but had no significant effect on other cytokines, indicating that immune stress was mainly caused by the induction of large amounts of TNF-α [41]. Furthermore, the results showed that the concentrations of serum IL-4 and IL-10 were significantly increased in broilers fed with FWB-IV, and the levels of pro-inflammatory cytokines TNF-α, IL-1β, IL-6, IL-8 and CRP were significantly decreased, which was consistent with the results of Kang et al. [30], and the mechanism may be that bioactive compounds in FWB, such as crude polysaccharides and total polyphenols inhibit the transcriptional expression of pro-inflammatory cytokines by downregulating the TLRs-NF-κB signal pathway and increase the levels of anti-inflammatory cytokines, further alleviate the inflammatory response caused by LPS-challenge [8,32]. The result indicated that the basal diet supplemented with FWB could alleviate inflammation, and the effect of wet FWB was better than dry FWB.

These results suggested that the nutritional properties, bioactive compounds and flavor properties of wheat bran were improved by solid-state fermentation, which may promote the growth performance and immune function of LPS-challenged broilers. Moreover, direct application of wet FWB can not only reduce drying cost and bioactive compounds, but also be more conducive to growth-promoting effect [7,8]. Therefore, FWB can be used as a functional feedstuff for growth immunomodulation in broilers.

## 5. Conclusions

Based on our result, we can conclude that the basal diet supplemented with 7% dry or wet FWB significantly improved the growth performance and strengthened serum immune performance of broilers, which effectively alleviated LPS-challenged damage. Moreover, the potential of wet FWB was better in broilers immunomodulation.

## Figures and Tables

**Table 1 animals-12-01100-t001:** Ingredients and nutrient levels of experiment diets.

Ingredients ^1^ (%)	Starter Diet (Day 1 to 21)			Grower Diet (Day 22 to 42)		
	Basal	Dry FWB	Wet FWB	Basal	Dry FWB	Wet FWB
Corn	57.99	50.22	50.22	60.69	55.16	55.16
Soybean meal	30.00	29.47	29.47	28.41	23.40	23.40
Corn protein flour	5.21	4.89	4.89	4.00	6.00	6.00
Dry FWB	−	7.00	−	−	7.00	−
Wet FWB ^2^	−	−	7.00	−	−	7.00
Soybean oil	3.00	4.57	4.57	3.50	4.99	4.99
CaHPO_4_	1.50	1.50	1.50	1.30	1.30	1.30
Limestone	1.20	1.20	1.20	1.20	1.20	1.20
Salt	0.30	0.30	0.30	0.30	0.30	0.30
Choline chloride	0.10	0.15	0.15	0.10	0.10	0.10
L-lysine HCL	0.20	0.20	0.20	0.10	0.15	0.15
DL-methionine	0.20	0.20	0.20	0.10	0.10	0.10
L-threonine	0.05	0.05	0.05	0.05	0.05	0.05
Vitamin premix ^3^	0.05	0.05	0.05	0.05	0.05	0.05
Mineral premix ^4^	0.20	0.20	0.20	0.20	0.20	0.20
Total	100	100	100	100	100	100
Nutrient levels ^5^						
ME (kcal/kg)	3056	3056	3056	3100	3100	3100
Crude protein (%)	21.50	21.50	21.55	20.26	20.00	20.00
Calcium (%)	1.07	1.07	1.07	1.01	0.97	0.99
Total phosphorus (%)	0.70	0.75	0.75	0.65	0.66	0.68
Available phosphorus (%)	0.47	0.49	0.48	0.43	0.41	0.43
Lysine (%)	1.27	1.26	1.26	1.12	1.07	1.08
Methionine (%)	0.54	0.54	0.54	0.42	0.43	0.43
Threonine (%)	0.85	0.84	0.84	0.80	0.79	0.78

^1^ Abbreviation: Dry FWB: dry fermented wheat bran; Wet FWB: wet fermented wheat bran; CaHPO_4_: calcium hydrogen phosphate; ME: metabolizable energy. ^2^ The moisture content of wet FWB was 30%, and the moisture content by adding 8.8% wet FWB was equivalent to 7% dry FWB dry matter content (DM basis). ^3^ The vitamin premix provided the following per kg of diets: vitamin A, 9500 IU; vitamin D3, 62.5 μg; vitamin K3, 2.65 mg; vitamin B12, 0.025 mg; vitamin B2, 6 mg; vitamin E, 30 IU; biotin, 0.0325 mg; folic acid, 1.25 mg; pantothenic acid, 12 mg; nicotinic acid, 50 mg. ^4^ The mineral premix provided the following per kg of diets: copper, 8 mg; zinc, 75 mg; iron, 80 mg; manganese, 100 mg; selenium, 0.15 mg; iodine, 0.35 mg. ^5^ Nutritional levels: ME was a calculated value, while crude protein, calcium and phosphorus were measured value.

**Table 2 animals-12-01100-t002:** Routine nutrient of wheat bran and fermented wheat bran ^1^.

Item ^2^	WB	FWB	*p*-Value
Dry matter, %	90.03	94.45	−
Crude protein, % DM	14.94 ± 0.06 ^a^	18.02 ± 1.23 ^b^	0.005
Acid-soluble protein, % DM	1.48 ± 0.01 ^a^	5.08 ± 1.84 ^b^	0.005
Crude fat, % DM	3.78 ± 0.06 ^a^	2.40 ± 0.39 ^b^	0.001
Crude fiber, % DM	16.01 ± 0.01 ^a^	14.65 ± 0.68 ^b^	0.044
Gross energy, MJ/kg	8.15 ± 0.01 ^a^	8.89 ± 0.41 ^a^	0.019

^a,b^ Means within the same rows without the same superscript letter were significantly different (*p* < 0.05). ^1^ Values were expressed as mean ± SD of 4 independent experiments (*n* = 4); compared on a dry matter basis. ^2^ Abbreviation: WB: wheat bran; FWB: fermented wheat bran.

**Table 3 animals-12-01100-t003:** Changes of bioactive compounds content in wheat bran and fermented wheat bran ^1^.

Item ^2^	WB	FWB	*p*-Value
Total phenolics (mg/g)	3.27 ± 0.01 ^a^	10.07 ± 2.57 ^b^	<0.001
Crude polysaccharides (mg/g)	9.02 ± 0.10	16.24 ± 12.24	0.189
Mannan (%)	0.23 ± 0.01 ^a^	0.49 ± 0.13 ^b^	0.024
Total dietary fiber (%)	31.27 ± 0.18 ^a^	49.65 ± 2.08 ^b^	<0.001
Soluble dietary fiber (%)	1.11 ± 0.08 ^a^	2.57 ± 0.87 ^b^	0.030

^a,b^ Means within the same rows without the same superscript letter were significantly different (*p* < 0.05). ^1^ Values were expressed as mean ± SD of 4 independent experiments (*n* = 4); compared on a dry matter basis. ^2^ Abbreviation: WB: wheat bran; FWB: fermented wheat bran.

**Table 4 animals-12-01100-t004:** Effect of fermented wheat bran on the growth performance of broilers challenged with lipopolysaccharide ^1^.

Item ^2^	Treatment Groups						SEM ^3^	*p*-Value		
	NC	PC	FWB-I	FWB-II	FWB-III	FWB-IV		Diet	LPS	Interaction
1 to 16 ^d^										
ADG (g/bird)	20.78 ^a^	21.78 ^ac^	23.75 ^b^	23.31 ^b^	22.90 ^bc^	23.58 ^b^	0.247	<0.001	−	−
ADFI (g/bird)	28.19 ^a^	29.31 ^a^	31.89 ^b^	32.71 ^b^	32.96 ^b^	32.55 ^b^	0.434	0.001	−	−
FCR	1.35	1.34	1.34	1.40	1.43	1.38	0.011	0.089	−	−
17 to 21 ^d^										
ADG (g/bird)	28.10 bd	23.33 ^a^	27.75 ^bc^	24.88 ^acd^	31.05 ^b^	26.32 ^acd^	0.655	0.075	0.001	0.700
ADFI (g/bird)	42.26 ^bd^	38.90 ^b^	44.26 ^cd^	41.30 ^bd^	49.93 ^a^	43.28 ^bd^	0.834	0.002	0.002	0.429
FCR	1.51 ^a^	1.67 ^bd^	1.61 ^acd^	1.66 ^cd^	1.61 ^ad^	1.64 ^ad^	0.018	0.526	0.034	0.311
22 to 42 ^d^										
ADG (g/bird)	54.62	55.06	55.42	56.10	57.81	55.86	0.463	0.129	0.657	0.110
ADFI (g/bird)	88.76 ^a^	93.08 ^ab^	96.56 ^bd^	96.01 ^bd^	102.43 ^c^	98.81 ^cd^	0.996	<0.001	0.971	0.098
FCR	1.62 ^a^	1.69 ab	1.74 bc	1.65 ^a^	1.77 ^cd^	1.77 ^bd^	0.014	0.002	0.716	0.048

^a–d^ Means within the same rows without the same superscript letter were significantly different (*p* < 0.05). ^1^ Values were provided as the means of 6 replicates (7 birds/replicate) in each control and treatment group (*n* = 6). ^2^ Abbreviation: NC: basal diet + sterile saline; PC: basal diet + LPS; FWB-I: 7% dry FWB + sterile saline; FWB-II: 7% dry FWB + LPS; FWB-III: 7% dry FWB + sterile saline; FWB-IV: 7% dry FWB + LPS; ADG: average daily gain; ADFI: average daily feed intake; FCR: feed conversion rate. ^3^ SEM: standard error of means.

**Table 5 animals-12-01100-t005:** Effect of fermented wheat bran on intestinal morphology of broilers challenged with lipopolysaccharide ^1^.

Item ^2^	Treatment Groups						SEM ^3^	*p*-Value		
	NC	PC	FWB-I	FWB-II	FWB-III	FWB-IV		Diet	LPS	Interaction
Day 21										
Jejunum										
Vh (μm)	668.16	553.56	813.86	767.01	747.41	873.60	42.909	0.126	0.890	0.496
Cd (μm)	149.01	150.17	139.24	137.77	122.68	154.85	6.645	0.762	0.454	0.557
V/C	4.74	3.63	6.71	6.64	6.51	5.64	0.530	0.161	0.528	0.917
Ileum										
Vh (μm)	334.73 ^a^	373.23 ^ab^	442.81 ^ac^	549.18 ^c^	479.48 ^bc^	512.07 ^c^	21.946	0.006	0.126	0.673
Cd (μm)	100.87	120.96	97.33	100.94	90.89	99.32	4.039	0.268	0.195	0.694
V/C	3.33 ^a^	3.09 ^a^	4.68 ^b^	5.76 ^b^	5.35 ^b^	5.04 ^b^	0.237	<0.001	0.596	0.172
Day 42										
Jejunum										
Vh (μm)	1050.84	898.63	736.3	928.83	835.47	854.6	41.820	0.320	0.814	0.259
Cd (μm)	123.77	155.59	97.02	129.13	115.94	103.70	6.466	0.101	0.161	0.237
V/C	8.89	6.08	7.77	7.26	7.30	8.51	0.395	0.888	0.38	0.134
Ileum										
Vh (μm)	643.73	728.28	551.83	719.31	674.65	632.84	27.175	0.749	0.212	0.307
Cd (μm)	99.04	96.58	90.36	106.52	109.39	95.81	4.271	0.895	0.997	0.409
V/C	6.44	7.62	6.18	6.82	6.38	6.18	0.214	0.572	0.095	0.767

^a–c^ Means within the same rows without the same superscript letter were significantly different (*p* < 0.05). ^1^ Values are provided as the means of 6 replicates (7 birds/replicate) in each control and treatment group (*n* = 6). ^2^ Abbreviation: NC: basal diet + sterile saline; PC: basal diet + LPS; FWB-I: 7% dry FWB + sterile saline; FWB-II: 7% dry FWB + LPS; FWB-III: 7% dry FWB + sterile saline; FWB-IV: 7% dry FWB + LPS; Vh: villus height; Cd: crypt depth; V/C: villus height to crypt depth ratio. ^3^ SEM: standard error of means.

**Table 6 animals-12-01100-t006:** Effect of fermented wheat bran on cecum chyme microflora of broilers challenged with lipopolysaccharide ^1^.

Item ^2^ (log cfu/g)	Treatment Groups						SEM ^3^	*p*-Value		
	NC	PC	FWB-I	FWB-II	FWB-III	FWB-IV		Diet	LPS	Interaction
Day 21										
*Escherichia coli*	5.65 ^a^	6.82 ^bc^	6.04 ^ac^	7.19 ^b^	5.78 ^a^	6.29 ^c^	0.149	0.155	0.001	0.458
*Lactobacillus*	7.16	7.33	7.22	7.03	7.07	6.78	0.058	0.057	0.307	0.187
*Bifidobacterium*	7.49	6.98	6.70	7.08	6.82	6.50	0.100	0.050	0.416	0.135
Day 42										
*Escherichia coli*	5.99	6.09	6.43	6.55	6.24	6.44	0.082	0.086	0.389	0.962
*Lactobacillus*	7.65 ^a^	7.65 ^a^	7.29 ^ab^	7.11 ^b^	6.59 ^c^	8.05 ^d^	0.099	0.008	0.001	<0.001
*Bifidobacterium*	8.75 ^a^	9.04 ^a^	7.44 ^b^	7.34 ^b^	7.4 ^b^	7.35 ^b^	0.151	<0.001	0.77	0.552

^a–d^ Means within the same rows without the same superscript letter were significantly different (*p* < 0.05). ^1^ Values were provided as the means of 6 replicates (7 birds/replicate) in each control and treatment group (*n* = 6). ^2^ Abbreviation: NC: basal diet + sterile saline; PC: basal diet + LPS; FWB-I: 7% dry FWB + sterile saline; FWB-II: 7% dry FWB + LPS; FWB-III: 7% dry FWB + sterile saline; FWB-IV: 7% dry FWB + LPS. Represents the beneficial and harmful bacteria counts in the cecum content. ^3^ SEM: standard error of means.

**Table 7 animals-12-01100-t007:** Effect of fermented wheat bran on serum parameters of broilers challenged with lipopolysaccharide ^1^.

Item ^2^	Treatment Groups						SEM ^3^	*p*-Value		
	NC	PC	FWB-I	FWB-II	FWB-III	FWB-IV		Diet	LPS	Interaction
Day 21										
TP (g/L)	18.93 ^a^	19.40 ^a^	23.36 ^bc^	23.48 ^bc^	20.68 ^ac^	23.54 ^bc^	0.607	0.010	0.291	0.529
ALB (g/L)	9.57 ac	9.24 ^a^	11.36 ^bc^	10.96 ^ac^	10.15 ^ac^	10.46 ^ac^	0.278	0.043	0.794	0.834
GLB (g/L)	9.35 ^a^	10.16 ^ac^	12.00 ^bc^	12.52 ^bd^	10.52 ^acd^	13.07 ^b^	0.384	0.009	0.057	0.393
TC (mmol/L)	2.17	2.16	2.23	2.12	2.34	2.20	0.080	0.867	0.621	0.947
TG (mmol/L)	0.88	1.12	0.90	1.01	1.46	1.11	0.076	0.175	0.977	0.267
GLU (mmol/L)	9.01 ^ab^	7.62 ^a^	9.08 ^ab^	9.49 ^b^	9.90 ^b^	9.75 ^b^	0.257	0.048	0.434	0.305
AST (U/L)	164.65	157.51	186.30	187.48	186.54	188.72	4.77	0.228	0.892	0.902
ALT (U/L)	5.78	6.14	6.09	5.44	6.54	5.33	0.22	0.932	0.277	0.374
Day 42										
TP (g/L)	26.84	24.28	24.68	25.74	21.68	24.56	0.636	0.242	0.715	0.213
ALB (g/L)	10.86	10.52	10.91	11.16	10.46	10.70	0.248	0.774	0.932	0.877
GLB (g/L)	15.98	13.76	13.77	14.58	11.22	13.85	0.439	0.06	0.599	0.053
TC (mmol/L)	2.24	2.10	2.25	2.19	2.21	2.18	0.051	0.931	0.494	0.919
TG (mmol/L)	1.47	1.07	1.26	1.16	0.97	1.21	0.086	0.694	0.630	0.355
GLU (mmol/L)	10.13	10.65	10.48	11.00	9.41	10.41	0.163	0.096	0.032	0.750
AST (U/L)	219.03	246.41	227.00	253.52	222.42	196.06	7.000	0.163	0.498	0.191
ALT (U/L)	6.58 ^a^	4.84 ^b^	5.12 ^b^	6.82 ^a^	4.58 ^b^	4.68 ^b^	0.228	0.008	0.961	0.001

^a–d^ Means within the same rows without the same superscript letter were significantly different (*p* < 0.05). ^1^ Values were provided as the means of 6 replicates (7 birds/replicate) in each control and treatment group (*n* = 6). ^2^ Abbreviation: NC: basal diet + sterile saline; PC: basal diet + LPS; FWB-I: 7% dry FWB + sterile saline; FWB-II: 7% dry FWB + LPS; FWB-III: 7% dry FWB + sterile saline; FWB-IV: 7% dry FWB + LPS; TP: total protein; ALB: albumin; GLB: globulin; TC: total cholesterol; TG: triglyceride; GLU: glucose; AST: alanine aminotransferase; ALT: aspartate aminotransferase. ^3^ SEM: standard error of means.

**Table 8 animals-12-01100-t008:** Effect of fermented wheat bran on serum immunoglobulins of broilers challenged with lipopolysaccharide ^1^.

Item ^2^ (g/L)	Treatment Groups						SEM ^3^	*p*-Value		
	NC	PC	FWB-I	FWB-II	FWB-III	FWB-IV		Diet	LPS	Interaction
Day 21										
IgM	1.07	1.05	0.91	0.91	0.81	1.22	0.048	0.379	0.167	0.111
IgG	3.96 ^ab^	3.72 ^a^	3.47 ^a^	3.50 ^a^	3.49 ^a^	4.53 ^b^	0.111	0.088	0.152	0.025
IgA	2.63 ^a^	2.07 ^ab^	2.08 ^ab^	2.25 ^ab^	1.78 ^b^	2.53 ^a^	0.091	0.552	0.480	0.012
Day 42										
IgM	0.84	0.76	0.78	0.89	0.77	0.96	0.032	0.689	0.274	0.239
IgG	3.35	3.18	3.24	3.48	3.42	3.28	0.058	0.780	0.812	0.324
IgA	1.86	1.92	1.79	1.88	1.76	2.08	0.076	0.908	0.335	0.767

^a,b^ Means within the same rows without the same superscript letter were significantly different (*p* < 0.05). ^1^ Values were provided as the means of 6 replicates (7 birds/replicate) in each control and treatment group (*n* = 6). ^2^ Abbreviation: NC: basal diet + sterile saline; PC: basal diet + LPS; FWB-I: 7% dry FWB + sterile saline; FWB-II: 7% dry FWB + LPS; FWB-III: 7% dry FWB + sterile saline; FWB-IV: 7% dry FWB + LPS; IgM: Immunoglobulin M; IgG: Immunoglobulin G; IgA: Immunoglobulin A. ^3^ SEM: standard error of means.

**Table 9 animals-12-01100-t009:** Effect of fermented wheat bran on serum cytokine of broilers challenged with lipopolysaccharide ^1^.

Item ^2^	Treatment Groups						SEM ^3^	*p*-Value		
	NC	PC	FWB-I	FWB-II	FWB-III	FWB-IV		Diet	LPS	Interaction
Day 21										
TNF-*α* (pg/mL)	37.22 ^a^	45.19 ^b^	50.08 ^bd^	53.65 ^cd^	54.63 ^cd^	56.51 ^cd^	1.562	<0.001	0.046	0.493
IL-4 (pg/mL)	9.45 ^bd^	8.97 ^bcd^	8.44 ^ad^	8.04 ^ac^	7.57 ^a^	9.98 ^b^	0.225	0.117	0.178	0.005
IL-6 (pg/mL)	99.49 ^a^	114.56 ^b^	117.47 ^b^	118.93 ^b^	119.70 ^b^	100.46 ^a^	1.972	0.005	0.732	<0.001
IL-8 (pg/mL)	34.00 ^ab^	35.28 ^b^	37.07 ^bc^	41.73 ^c^	35.61 ^b^	29.44 ^a^	0.963	0.003	0.961	0.021
IL-10 (pg/mL)	18.87 ^bd^	18.02 ^bd^	16.27 ^cd^	14.84 ^ac^	13.40 ^a^	19.79 ^b^	0.533	0.012	0.072	<0.001
IL-1*β* (pg/mL)	17.76 ^ab^	18.57 ^b^	19.76 ^b^	22.50 ^c^	23.13 ^c^	16.25 ^a^	0.534	0.002	0.078	<0.001
CRP (mg/L)	3.07 ^b^	3.86 ^c^	4.02 ^cd^	4.34 ^cd^	4.47 ^d^	2.48 ^a^	0.147	0.001	0.049	<0.001
Day 42										
TNF-*α* (pg/mL)	48.32 ^ab^	52.70 ^bc^	55.86 ^c^	65.93 ^d^	74.76 ^d^	45.98 ^a^	2.025	<0.001	0.009	<0.001
IL-4 (pg/mL)	8.34 ^b^	7.26 ^a^	7.15 ^a^	7.08 ^a^	6.54 ^a^	8.78 ^b^	0.186	0.102	0.178	<0.001
IL-6 (pg/mL)	114.56 ^a^	122.28 ^a^	124.74 ^a^	142.09 ^b^	148.41 ^b^	108.94 ^a^	3.309	0.033	0.291	<0.001
IL-8 (pg/mL)	38.63 ^ab^	39.74 ^bc^	43.63 ^bc^	46.92 ^cd^	53.45 ^d^	30.87 ^a^	5.690	0.109	0.013	<0.001
IL-10 (pg/mL)	15.97 ^b^	15.36 ^b^	14.98 ^b^	13.88 ^ab^	11.20 ^a^	17.37 ^b^	0.580	0.479	0.153	0.011
IL-1*β* (pg/mL)	21.39 ^ab^	22.98 ^abc^	24.33 ^bd^	25.34 ^cd^	20.30 ^a^	24.70 ^bd^	0.561	0.077	0.027	0.342
CRP (mg/L)	4.10 ^a^	4.64 ^b^	4.78 ^b^	4.90 ^b^	4.98 ^b^	3.85 ^a^	0.096	0.008	0.223	<0.001

^a–d^ Means within the same rows without the same superscript letter were significantly different (*p* < 0.05). ^1^ Values were provided as the means of 6 replicates (7 birds/replicate) in each control and treatment group (*n* = 6). ^2^ Abbreviation: NC: basal diet + sterile saline; PC: basal diet + LPS ; FWB-I: 7% dry FWB + sterile saline; FWB-II: 7% dry FWB + LPS; FWB-III: 7% dry FWB + sterile saline; FWB-IV: 7% dry FWB + LPS; TNF-α: tumor necrosis factor; IL-2: interleukin-2; IL-4: interleukin-4; IL-6: interleukin-6; IL-8: interleukin-8. IL-10: interleukin-10. IL-13: interleukin-13; IL-1β: interleukin-1β; CRP: acute C-reactive protein. ^3^ SEM: standard error of means.

## Data Availability

All data sets collected and analyzed during the current study are available from the corresponding author by request.

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
