# Peer review of "Effects of Solid-State Fermented Wheat Bran on Growth Performance, Immune Function, Intestinal Morphology and Microflora in Lipopolysaccharide-Challenged Broiler Chickens"

_animals, 2022, doi:10.3390/ani12091100_

Round 1

Reviewer 1 Report

In this research the authors tried to detect the bioactive components in solid-state fermentation wheat bran and compare the effects of dry and wet fermentated wheat bran on productivity in LPS-challenged broiler chickens.

The topic is original and the manuscript falls within Animals scopes carefully.

There is not any similar report on LPS-challenged broiler.

The methodology is prepared carefully and I think it is ok in present form.

The conclusions  are consistent with the evidence and arguments presented and they address the main question posed.

The references are enough and appropriately cited in the text.

The manuscript was properly conducted and findings reported are important for poultry production and nutrition. The paper contains valuable data. The authors investigated an interesting topic and the objective of the paper is of worldwide interest and fits well within the overall scope of the journal. Results were properly reported and the findings have been accurately discussed and compared with other published papers.

So, based on my opinion the manuscript merits the acceptance.

Reviewer 2 Report

The study presents interesting results on the use of an alternative feed ingredient (fermented wheat bran) in broiler chicken production. Despite the relevance of this study, there exist a few inaccuracies related to result interpretations shown below-

  • Line 29-30- The statement- “Compared with NC, dry or wet FWB increased…” is only true for ADFI for the d22-42 period. Please revise accordingly.
  • Line 32- “…decreased the jejunum crypt depth (p < 0.05)..”, according to Table 5, values for crypt depth were not significant. Please revise accordingly.
  • To ensure clarity in the result presented, results for dry FWB and wet FWB might need to be presented separately.
  • Line 32- “and the Escherichia coli counts of…” Is an increase or decrease reported here? In any case, this is not valid according to Table 6, p=0.086 is reported.
  • Line 34- “…albumin, globulin, glucose, and alanine aminotransferase on day 21 (p < 0.05)” statement not true for ALB and ALT according to Table 7. While values for ALT reported are not statistically significant, those for ALB are statistically comparable to the NC values.
  • Line 35-36- “…decreased the levels of serum total protein, aspartate aminotransferase…” statement not true for total protein and aspartate aminotransferase according to Table 7. Please revise accordingly.
  • Line 36-37- “…FWB-IV increased the levels of serum immunoglobulin A…” statement only true for immunoglobulin G according to table 8.
  • Line 37-40- “Compared with PC and FWB-II, FWB-IV decreased the levels…” results of day 21 might need to be presented separately from day 42, to ensure clarity. The quoted statement might not be completely true for IL-6, IL-8, IL-1β, C-reactive protein (increased), TNF-α (increased); IL-4 and IL-10 are only true for FWB-IV. Please revise accordingly.
  • Line 173-179- Clarity needed. How many birds were sampled per treatment? Was the feed restriction for all birds in the experiment or for only birds to be sampled?
  • Line 185-192- two sets of method were provided for serum biochemical analysis; please clarify
  • Line 199- provide information on tissue morphometric processing i.e., staining, cutting, etc.
  • Line 232- Please rephrase, there was no change in crude polysaccharide according to the P-value presented in Table 3. Also, if p-value is not <0.001, as is the case for “soluble dietary fiber” in table 3, state the exact P value
  • Line 241-243- “Compared with basal diet groups, dry and wet FWB groups significantly…” during which period? statement not entirely true; during d17-21, only FWB-III (7% wet FWB + sterile saline) recorded a significant increase in ADFI compared to the NC treatment; all other parameters were statistically similar for all other treatments. Only during d22-42 did both dry and wet FWB increase ADFI compared to the NC group.
  • Line 244-246 “…dry and wet FWB groups significantly increased ADG and ADFI…” Not true for ADG for both periods and only true for ADFI in FWB-III treatment during d17-21.
  • Line 271-273- “…compared with the basal diet groups, wet FWB groups significantly decreased…” only true for Please revise for correctness.
  • Line 282-283- “Compared with the NC, dry and wet FWB groups significantly increased TP, ALB, GLB, GLU, and AST levels (p < 0.05).” most of the parameters listed share similar statistical levels as the NC group according to table 7.
  • Line 284-285- “During the later period (days 22 to 42), GLU was content significantly increased (p < 0.05)..” in what treatment and compared to what treatment?
  • Line 289-290 “…but the wet FWB group significantly decreased GLU, AST, and ALT levels (p < 0.05)” only true for ALT.
  • Line 291- Table 7, p value of 0.042 is reported for AST for day 21, yet the means are not separated by Duncan’s multiple range test.
  • Line 301-304- “However, compared to the PC and FWB-II, the FWB-IV..” please revise for correctness, stating the period of comparison, the basis of comparison while paying close attention to superscripts shared by the displayed mean values on the presented table.
  • Line 299-300- “In the PC and FWB-II, the pro-inflammatory 299 cytokines IL-6, IL-1β…” what period and what is the basis of comparison?
  • Line 342-crude polysaccharide was not significantly increased; please rephrase for correctness
  • Line 366-368- “The current results showed that the E. coli counts of cecum digesta were significantly…” This is not true for all FWB groups. It might be better to state the specific FWB group being referred to here, as well as the basis of comparison.

Minor comments

  • Line 18-19- “…and the feeding effect of wet FWB was better in…” Please rephrase for clarity. “wet FWB improved the immune profile of broiler chickens…” might be a better alternative.
  • Line 23- “broiler" should be "broiler chickens”
  • Line 26- “containing 6 replicates….” should be “containing 6 replicates replicate cages”
  • Line 40- “…supplement of 7%...” should be …supplementing 7% …”
  • Line 49- “LPS challenging “should be “LPS challenge”
  • Line 50- there should be a comma after “inflammatory response”
  • Line 53- “more” should be “most”
  • Line 57-61- This seems like a lengthy sentence, could be split into multiple simple sentences.
  • Line 73- More background information on solid-state fermentation could be provided- the process and benefits, including the pros and cons of wet vs. dry solid-state fermented products.
  • Line 75-77- This also seems like another lengthy sentence, which could be split into multiple simple sentences.
  • Line 79- “modulated the intestine microbiota” positively or negatively? Please clarify.
  • Line 85- “is need further study” should be “needs further study”
  • Line 86- “fermentation” should be “fermented”
  • Line 137- delete “were compiled” should be “complied”
  • Line 146- “with containing 6 replicates of 7 broilers” should be “containing 7 broiler chickens/cage and 6 replicate cages/treatment”
  • Line 148- how many ml sterile saline solution was injected?
  • Line 164- Table 1, are the ingredients presented on an as-fed or dry matter basis? Also, please define all ingredients and chemical formula presented in abbreviations in the footnotes
  • Line 182- define EP
  • Line 227- define the abbreviation TCA-N
  • Line 249- Table can start on a new page; all treatments abbreviation should also be defined in the footnotes. Although they may have been defined in the main article, tables are considered independently.
  • Line 256-257- “…dry and wet FWB groups significantly increased ADG and ADFI…” when? state the period or phase referred to here
  • Line 267-269- “…but had no significant effect on the coli counts (p > 0.05).” state the study period referred to here.
  • Line 271- “…. but Lactobacillus counts were significantly increased (p < 0.05)” compared to what? Please state the basis of comparison
  • Line 354-358- please edit for clarity.
  • Line 372- could more information on why the LPS challenge did not cause a change in IgG and IgM levels be provided.
  • Line 394-400 -To ensure logical flow and comprehension, can the results be discussed in the same order the table of results are presented? This will mean the results on serum parameters will be discussed before those on serum immunoglobulins and serum cytokines.

Reviewer 3 Report

Dear 

Editor

Animals

The manuscript is interesting, well written and discussed.

I would like to thank the Editorial Staff of Animals for entrusting me with the review of the Article Manuscript ID: Animals -1648173 entitled: Title: Effects of solid-state fermented wheat bran on growth performance, immune function, intestinal morphology and microflora in lipopolysaccharide-challenged broiler chickens

Authors; Jishan An , Jingjing Shi , Kuanbo Liu , Aike Li , Beibei He , Yu Wang , Tao Duan , Yongwei Wang, Jianhua He

One way to deal with the rising cost of feed due to the growing demand for grains could be to develop alternative materials to the main feeds such as wheat bran. Annually, more than 770 million tons of wheat are produced worldwide, and a large amount of wheat is produced when processed into flour. However, high lignocellulosic content and low nutrition value (kcal/kg of ME) makes it unsuitable as supply to monogastric animals. Furthermore, nonstarch polysaccharides contained in lignocellulosic tend to act as antinutritional compounds that could inhibit digestibility, causing pathogen proliferation in the gastrointestinal tract and inducing gut inflammation. Compared with the submerged fermentation, solid-state fermentation is more efficient and cost effective in producing bioactive compounds.

There are several major issues that need to be addressed. I would like to point out that the article is a continuation of other studies in this field on broilers.

The objectives are clear and are to detect bioactive components in solid-state fermentation wheat bran and compare the effects of dry and wet FBW on growth performance, resistance function, intestinal morphology and microflora in broiler chickens. 

Throughout the text, attention should be paid to the citation of items, and the indication of the names of tables and figures - the text is not bold.

Introduction: It is written correctly and analyzes the issue.

Materials and methods:

P: 89 There is no declaration of permission to use animals in these studies. This is all the more required in the case of grant research or national programs. There is only a declaration of national animal testing. I am asking for a deeper explanation - ask for the license number. 

P: 90 Was there a utilization procedure for all microorganisms to be used in these studies?

P: 99-100 Why such a time of 15 minutes. Why is the temperature 37 ° C

P: 108 Fermentation temperature and time. Why 3 days and temperature 23-33 0C?

P:110 Did these samples have the optimal storage temperature? . Why should 4 0C, be -20 0C?

P: 180 Why is spin 3000?

P: 212 A mathematical model should be provided.

P: 227 Table 2. Crude, fat content has dropped, not increased.

P:272-273 Unclear statement

The number of animals experimental is small. I suggest that the test should be repeated on more animals.

The Discussion of the results is consistent with the obtained research results.

I suggest using additional literature for Material and methods, Introduction and Discussion.

Summing up, I can say that due to the interesting results obtained, sometimes controversial and comparing many nutritional groups and parameters, the work should be accepted for publication after taking into account the reviewer's suggestions.

After making corrections and supplementing, the article can be accepted for publication.

Round 2

Reviewer 2 Report

Well done with the responses!  

To ensure clarity and consistency, can the P-values written within the manuscript text come immediately after the verb 'increase" or "decrease". This may be revised throughout the manuscript. 

Author Response

Thanks for your advice. Please see the attachment
